# Exploring Staff–Student Partnership in Curriculum Design

**Fiona Alison Boyle** [1,*] , **Fiona M. Buchanan** [1] , **Dan Ritchie** [1] **and Kelum A. A. Gamage** [2]

1   School of Law, University of Cumbria, Carlisle CA1 2HH, UK; fiona.buchanan@cumbria.ac.uk (F.M.B.); dan.ritchie@cumbria.ac.uk (D.R.)
2   James Watt School of Engineering, University of Glasgow, Glasgow G12 8QQ, UK; kelum.gamage@glasgow.ac.uk
*   Correspondence: fiona.boyle@cumbria.ac.uk

**Abstract:** Collaboration between staff and students for improved curriculum design is presented as highly beneficial in many contexts. In particular, Education for Sustainable Development (ESD) is seen as facilitating co-design and collaboration. However, students tend not to be actively involved in curriculum change, especially in whole programme design or review. Furthermore, few studies consider partnership with students in the context of ESD inclusion, which was the focus for this collaboration. The aim of this study was to explore staff and student perceptions of participating in a collaborative approach to the review and development of the undergraduate law curriculum in a university in the UK. A developmental evaluation using mixed methods was used to study the expectations and experiences of students and staff through a process of curriculum review and change to incorporate ESD. Our findings demonstrate the rewards of a collaborative process incorporating diverse perspectives. However, they also highlight barriers for students and staff, including perceptions of risk associated with student involvement in the process, and we offer reflections on navigating these potential risks.

**Keywords:** students as partners; curriculum design; co-creation; education for sustainable development; education innovation; developmental evaluation

## 1. Introduction

This article describes an example of collaborative curriculum change using the Students as Partners (SaP) model to update an undergraduate programme and incorporate Education for Sustainable Development (ESD). Recently in this journal, Kioupi and Voulvoulis, have advocated for participatory approaches to curriculum development [1]. The focus of this article is on one such approach, the SaP process; however, we see context and focus as entwined. Therefore, we set out below the connections between this process and contextual factors, notably embedding ESD in the curriculum in question, an undergraduate law programme in the UK. To understand if and how SaP might be used to enhance student engagement, this project invited students to be part of a periodic review of the law curriculum where the focus was on incorporation of ESD. This project thus considers SaP use in the context of legal education and SaP theory is described in more detail in Section 3 below.

SaP is a growing area of scholarship and practice promoting meaningful student engagement in higher education at several levels. These levels can include classroom activities, assessment design, research, and the wider curriculum. The example presented involved a formal institutional review of the whole curriculum where one of the key objectives was to incorporate ESD within the programme. The research aimed to gather data regarding participant understandings and aspirations and to identify factors that either inhibited or promoted the use of SaP in this context. We found that there were significant benefits for the outcome and rewards resulting from a collaborative process incorporating diverse perspectives. These included an outcome that was informed by student experience and direct input. Rewards for those involved also included for students, satisfaction

through contributing to positive change and for staff, an enhanced understanding of student views and needs. However, the data also highlight potential barriers for students and staff, including perceptions of risk associated with student involvement in the process, and we offer our reflections on navigating these potential risks.

SaP is built upon ideas about active learning and positions students as equal contributors to teaching, learning, assessment, and research practices. This means that collaborative approaches are central to the concept of SaP. Likewise, ESD is presented in policy as a means to "facilitate a culture of co-design and collaboration" [2]. This means that both policy guidance and information regarding student preferences appear to support a collaborative or partnership approach to curriculum development in this area. Furthermore, collaborative competency is one of the eight UNESCO key competencies for sustainability [3]. SaP is presented as a form of student engagement and aligns closely with the concept of "student voice" when used in the context of curricula co-creation. SaP seeks to enhance individual and collective experiences of higher education (HE) by incorporating student knowledge of what it means to be a learner in a particular context. Overall, SaP offers a framework within which to reconceptualise traditional approaches and relationships. A stated aim is to empower students to work with staff in any relevant area of HE to enhance experience, learning, and policy [2]. Thus, it positions students as active and agentic rather than as passive recipients or consumers of education [4]. Despite increasing interest in SaP, its use in curriculum design has been less well developed [5] and with notable exceptions [6], there are as yet, few studies considering SaP in the context of ESD. Furthermore, recent research considering curricula change and the extent of co-creation in the curriculum in the UK [7] indicates that while collaboration with students is often seen as an aspiration, it is not often enacted in practice.

Matthews et al. [8] have noted that the SaP literature often emphasises the benefits of its use, while "messiness" is occluded. This article therefore attempts to provide an honest account of navigating both the risks and rewards of a partnership approach to curriculum review. To achieve this, it describes an example in which SaP was used to inform the process of a formal review of the undergraduate law curriculum where one of the objectives was to incorporate ESD. It begins by situating the discussion within the current position of undergraduate legal education in the UK, including recent changes affecting curriculum design and content specifically as they relate to sustainability, addressing the potential congruence between the SaP process and sustainability. The article then reviews the SaP literature, its connection to the concept of student engagement, and its relevance to student involvement and co-creation within undergraduate legal education. It sets out the research design used to enable readers to consider both similarities and differences when compared to their own context. A discussion of the results and tentative lessons learned are then considered. While the limitations of this small-scale study are acknowledged, it is also argued that others might find value in understanding the knowledge gained from this experience.

## 2. Context and Background

The institutional context for this research was a new university in the northwest of England. The law school is small, comprising, at the relevant time, six staff and approximately fifty undergraduate students. Students are often the first in their family to attend university. A high proportion of students are commuter students and the student cohort at the time of the study was 78% female. The programme is generally viewed by students and staff in vocational terms, as a means to enter the legal profession, and while some students go on to alternative careers, many are employed within local legal service providers after graduation. The academic staff are 57% female and most have some background in professional legal practice. The institution operates using an academic departmental structure supported by professional service staff. Senior management staff tend to oversee specific areas such as student success, portfolio development, or learning, teaching, and assessment.

In England, the legal professions have historically exerted considerable control over undergraduate course content; this diminished to some extent in 2021 following a different path to qualification [9]. In addition to this loosening of professional bodily control, the revised UK subject benchmark for law [10] now incorporates a reference to the inclusion of sustainability in the law curriculum. These changes presented an opportunity to engage with the law curriculum in different ways [11] and it was considered timely to use a different review process that might facilitate contextual development of the curriculum away from previously high-level strategically controlled content [12].

ESD was taken as the key factor promoting inquiry and change to the curriculum. Sustainability was seen as of prime importance due to its local and global significance and the role of higher education (HE) in contributing to the achievement of the UN's Sustainable Development Goals. Local sustainability issues that were considered significant included the management of the local economy, including food production, energy infrastructure, and tourism. Globally, the UN 2020–30 Decade of Action highlights the role of HE in ESD and there is a growing community of interested parties in Higher Education for Research and Sustainable Development (HESD) covering all disciplinary areas. In this article, the definition of ESD used is that provided in UK guidance for HE [2], this is "the process of creating curriculum structures and subject relevant content to support and enact sustainable development" (p. 51). The aim was to take inspiration from the introductory remarks of one of the joint authors of this QAA guidance, Professor J. Longhurst, who states, "[i]f we as educators are serious about preparing our students for the future, we must embrace ESD and ensure that every graduate has not only the knowledge and skills but also the attributes that will enable them at least to cope and ideally thrive in the face of the multiple challenges they will face across their life course in the 21st century." [2]. David Orr, a seminal writer in the area of environment and education, famously stated in relation to the development of environmental crises that confront the world, "this is not the work of ignorant people. It is, rather, largely the result of work by people with BAs, BScs, LLBs, MBAs, and PhDs." [13]. This means that legal educators within HE have a potential part to play in ensuring that future LLB graduates have the knowledge and skills referred to by Prof. Longhurst. Existing disciplinary research has already pointed to the significance of sustainability for legal education and likened the concept of sustainability to higher-order goals such as democracy, justice, and the rule of law [14] which currently underpin the law curriculum.

However, despite the revised benchmark and guidance, there are potentially still significant obstacles. Graham [14] offers a detailed critique of legal education's taxonomy and anthropocentrism, while issues such as conservatism, narrow definitions of the curriculum, and bias towards vocationalism have also been highlighted as barriers [15]. Significant research regarding the integration of ESD in law curricula was reported by Lowther and Sellick [16]. In that study, students were involved in giving feedback on staff-designed curriculum change and the authors reflected on the difficulties of balancing traditional law content and ESD content, identifying a need for further research on ESD's significance to law students. Although there is no large-scale study considering this question in relation to law students as a group, a sustainability skills survey [17] showed that 79% of student respondents agreed that sustainable development is something all university courses should incorporate. This indicates that the use of more collaborative processes including students as co-designers may act to promote change in this area of curriculum development.

Early research regarding the incorporation of sustainability in the law curriculum [15] called for information and examples that examined the means by which sustainability could be addressed as part of curriculum review, also calling for models of implementation. The use of SaP can provide such a model because its utilisation is potentially beneficial and relevant to legal education for a number of reasons. Legal education has traditionally relied on the lecture format in which students are often passive recipients of information. However, this approach has been problematised as lacking in capacity for building a community of learners and encouraging more superficial engagement [16]. In addition, SaP

promotes inclusive and equitable learning environments which value diverse perspectives. This is another area in which it has been suggested that legal education might work to further enhance the flourishing of all students [18]. Thus, SaP has the potential to enhance legal education by fostering collaborative and inclusive HE environments which better prepare students for future legal practice.

This article will therefore provide a description and reflections on one example of ESD incorporation using the SaP model to facilitate co-design with students. Tun et al. [19] presented evidence of SaP use when integrating sustainable healthcare education, and this case study aims to present similar evidence in the context of legal education while noting the limitations regarding generalisability from one example. Thus, due to the centrality of law in "securing sustainable development and the wellbeing of future generations" [10], the objective in the example reported here was to embed ESD into the law curriculum in partnership with students. This was achieved in a way that accounted for staff expertise, student interest, and other academic quality requirements, capitalising on the potential congruence between SaP and ESD. Having outlined the rationale for the focus on ESD, the following section considers the relevant SaP literature and frameworks and situates these within the wider field of student engagement.

### 3. Theoretical Framework: Students as Partners as a Form of Collaborative Student Engagement

This section sets out a definition of student engagement and then situates and defines SaP as a collaborative form of engagement. It sets out some of SaP's key characteristics and explains how SaP theory was used in this research. After clarifying the relationship between student engagement and the concept of SaP, this section concludes by considering in more detail one of the key issues considered in the literature: the question of empowerment.

Despite the fact that student engagement is a broad, contested, and ambiguous field [5] with many dimensions [20], it is almost universally accepted that student engagement results in "positive outcomes of student success and development, including satisfaction, persistence, academic achievement and social engagement." (Trowler) [21]. The SaP literature can be seen as a subset of the student engagement literature [22]. It is an emerging area intersecting with research regarding student voice. SaP employs definitions that emphasise developmental collaboration and the most cited definition describes SaP as the following:

> "*a collaborative, reciprocal process through which all participants have the opportunity to contribute equally, although not necessarily in the same ways, to curricular or pedagogical conceptualization, decision making, implementation, investigation, or analysis*". [23]

SaP is seen as empowering for students and thus an alternative to consumerist models of HE [5]. It encompasses the concept of the co-creation of a curriculum [24–29]. Although students are central, partnership can be with other students, staff, administrators, alumni, or employers [30]. SaP in the context of curriculum design and review, as is described here, can be seen as inviting students behind the scenes of institutional processes.

The conceptual framework developed by Healey et al. [5], a version of which is set out below (Figure 1), represents SaP within the wider field of student engagement, setting out four overlapping domains with a learning community as a central feature. In addition, it depicts nine key features of successful engagement, including inclusivity, plurality, courage, and empowerment. This framework offers a means by which to map various contextual practices. The example described below and considered in this article falls within the area of curriculum design and pedagogic consultancy.

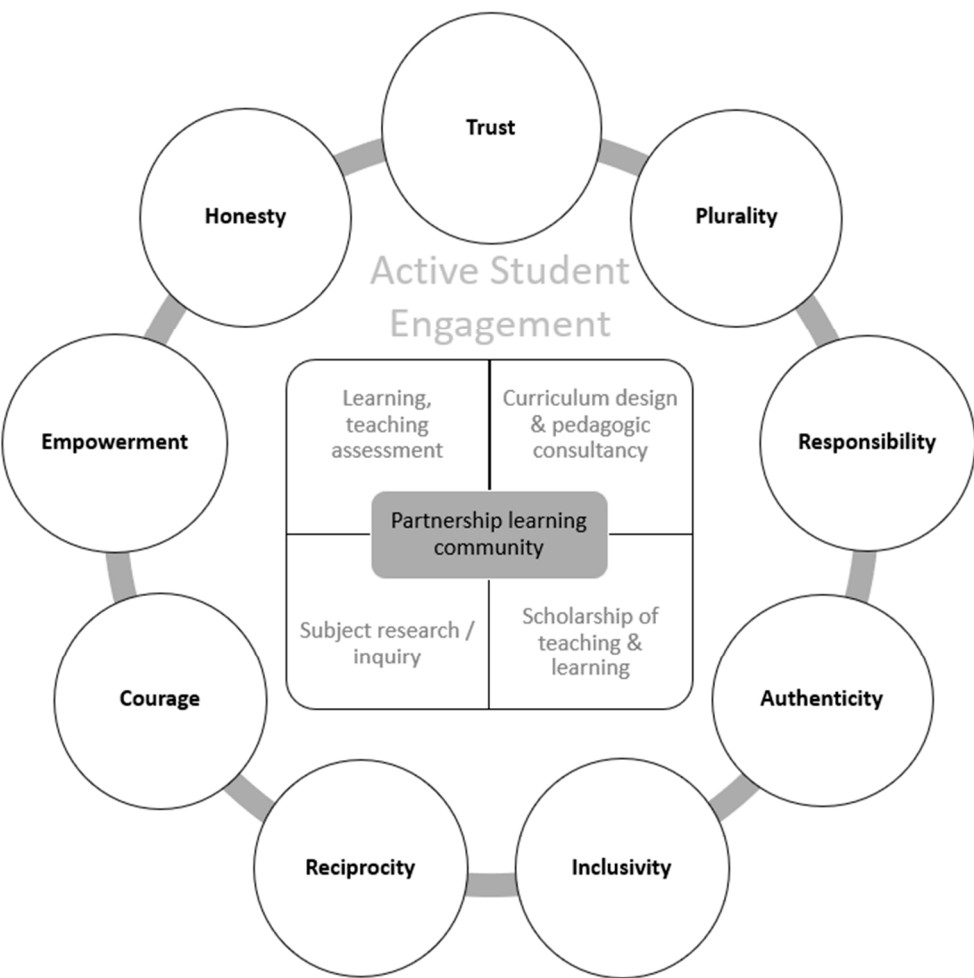

**Figure 1.** Students as partners model.

Reflecting the nine characteristics of SaP identified in Healey et al.'s model above, other research has suggested five propositions for good practice and working in "genuine" partnership [30]. These are the following:

1. To foster inclusivity;
2. To use dialogue and reflection to nurture power-sharing;
3. To accept uncertain outcomes;
4. To work ethically;
5. To aim towards transformative partnership.

It is clear that SaP is not necessarily an easy option. There are many challenges in moving beyond more passive methods, such as evaluation surveys, as a means of engagement. These include issues of power, scale, and context, staff and student vulnerability, questions of who is represented and who is not, the importance of language, resistance, risk, the time taken to create a partnership, and the level of perceived participation [25]. Healey et al. [5] added to this list of issues to consider, reward/recognition, transition, and the sustainability of the partnership. We will consider below in more detail the question of power imbalances and explain why, in this context, the notion of altering the balance of power between staff and students was seen as a risk rather than a reward of the process, by both staff and students.

Much of the SaP literature considers the concept of student power within the partnership and often presents power as on a continuum, from tutor-controlled curricula through curricula informed by feedback to a curriculum negotiated with students [31,32]. We aimed to create a negotiated curriculum incorporating ESD in a way that would involve compro-

mise, but also reflect the wishes of as many participants, students, and staff, as possible. Considering the continuum of student engagement in curricula, many mechanisms widely used to capture student voice can be categorised as feedback, relying as they do, on post hoc survey responses. For example, the UK National Student Survey claims to give "students a powerful collective voice" [33], but authors such as Velden et al. [34] note that overreliance on student surveys can create a significant barrier to student influence rather than an enhancement to it. Furthermore, survey fatigue and the negative impact of such data [35] are argued to impede change and disincentivise innovation whereas a more developmental, integrated use of staff and student data may allow for dynamic and responsive curricula adjustments [36].

When working with students as partners, imbalances in power can create barriers in activities and discourse. The literature on power is wide and varied, but within universities, social relationships between staff and students are an important element. Traditional hierarchies between students and staff are being reframed in relation to wider changes in higher education since the raising of student fees and marketisation of HE has brought about a relationship in which the power is with students as consumers to demand value for money [37,38]. These power dynamics may hamper collaborative working between academic staff and students. Indeed, the notion of the consumer student suggests a one-way relationship rather than a partnership. For example, Freeman [39] argues that student representatives are now part of HE consumerism, and partnerships are used by students to advance their commercial interests, rather than creating a more democratic decisionmaking culture. It has been noted by authors such as Williams [40] that the marketisation of higher education categorises students as complaining consumers, rather than giving them a proper outlet for influence. Instead, many processes operate to allow and encourage student complaints on issues such as service delivery, while excluding them from higher-level dialogue on issues such as teaching quality. Symonds, considering the relevance of power and context in partnership practice, notes that in UK post-1992 universities, there is an assumption of the "traditional power relationship. . .with undergraduate students who are less confident and thus, more likely to defer to the entrenched power of the academic teacher role, which conflicts heavily with the shared power of the partner role" [41]. This relationship will depend not only on the institution, but also on the programme of study and the culture within the department.

This article is based on experience within a small university department. The small class sizes (typically no more than 20 in a year group) allow the staff and students more time together on an individual basis and this helps foster relationships which are often open. While professional distances must be maintained, small groups facilitate relatively relaxed teaching and more time for interaction. Despite these contextual factors, and the achievement of what were considered by students and staff as the benefits of an excellent outcome based on an inclusive process, the potential pitfalls also generated concern. Both students and staff saw risks in a partnership approach. Therefore, after setting out some details regarding the research design, both the rewards and the perceived risks of partnership are considered in our discussion below.

## 4. Research Design

During the academic year of 2021–2022, a developmental evaluation using mixed methods was undertaken by one of the co-authors. This involved one cycle of curriculum review in which, together with a student survey, the qualitative perceptions of final-year law students, academic staff (including the co-authors), student union representatives, and professional and managerial staff were recorded using semi-structured interviews during three phases as illustrated in Figure 2. The questions guiding the research were the following:

- What are participant understandings and aspirations regarding working in partnership?
- What factors act to either promote or inhibit working in partnership?
- What are the implications for future practice and use of SaP in curriculum review?

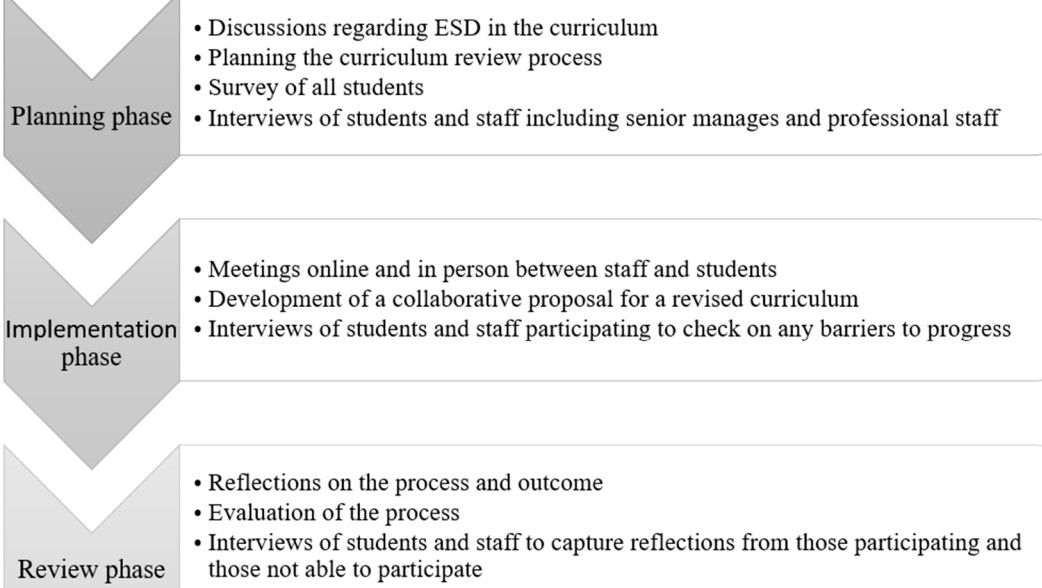

**Figure 2.** The three phases of the initial curriculum review cycle.

As we wished to explore the staff–student experiences of participating in a collaborative process of curriculum design, a mixed methods approach was chosen for this study. While quantitative survey data provided important baseline information regarding student understandings and aspirations prior to the review, participant values, beliefs, and reflections on the process were gathered using semi-structured interviews. Developmental evaluation is grounded in a philosophically pragmatic paradigm which tends to favour mixed methods [42]. In this case, comparison of quantitative and qualitative data was possible, but the dominant method was qualitative.

The process of change to be evaluated involved a staff–student partnership in curriculum review to embed ESD. Developmental evaluation is a recognised methodology which supports, inter alia, an innovation in the form of a system change. It is distinguished from other forms of evaluation as it allows for the monitoring of a developing situation as it is implemented, where the outcomes are not precisely specified. For this reason, it was considered to be appropriate in our research. Patton [42], the leading proponent of developmental evaluation, offers eight sensitising concepts which were applied throughout the process. These sensitising concepts require that, as far as is possible, developmental evaluation should adhere to the following:

*Be Co-created*—though led by one member of academic staff, the evaluation involved other staff and students. The co-creation of curricula using a Students as Partners approach working with all programme teaching staff therefore aligns with developmental evaluation.

*Be Considerate of Complexity*—the dynamic and complex environment of higher education [43–45] is recognised and its effects acknowledged.

*Use Systems Thinking*—consideration is given to relationships, connections, perspectives, and both intended and unintended consequences acknowledging temporal, spatial, and other contextual issues.

*Be Developmental*—this means that the change is evaluated as it occurs and there is often a cyclical element to the process of change, as in curriculum (re)design.

*Be Driven by Innovation*—the innovative change here was the partnering with students to incorporate ESD into the curriculum.

***Be Utilisation-focused***—in this case, the focus was on the identification of promoting and inhibiting factors regarding use of partnership in the context of curriculum review to inform future action.

***Be Rigorous***—using a mixture of qualitative and quantitative data as appropriate [45] and testing existing theory, here in relation to Students as Partners.

***Timely***—the findings, some of which are set out as *lessons learned* below, are available as needed and can be used to inform future action.

### 4.1. Methods

During the year, the project was constructed, specific actions were planned, and a series of four meetings took place (some online and some in person) culminating in a formal review meeting involving an external panel member, law academics, a student reviewer, and two final-year students. In order to facilitate free and open discussion among all participants, the meetings did not form part of data generation. Data collection was conducted separately, via a survey instrument (Appendix A) and semi-structured interviews. Prior to the review process, the survey was distributed to all programme students seeking views on their aspirations regarding the process and its potential outcomes. The survey gathered both quantitative and qualitative data. Because all programme students were surveyed, it was not possible to pilot the questions. However, in line with the developmental evaluation approach, the survey was produced by programme staff. Questions were developed with reference to relevant theories regarding understandings and aspirations related to staff–student partnership and used a Likert scale [46]. Interview questions were guided by the research questions and a developmental evaluation inquiry framework focused on participant beliefs, knowledge, and action [45]. An outline of the participants, methods, data, and forms of analysis used is provided in Table 1, along with an indication of the phases during which the data was gathered. The quantitative survey data were not intended to be generalisable and were analysed using descriptive statistics. The interview data were analysed using Reflexive Thematic Analysis [47] and guided by the research questions. This method of qualitative thematic analysis was chosen because it facilitates the telling of a multi-faceted story across the whole dataset. It is worth noting that the number of transcripts does not correspond to the number of participants because firstly, some participants were interviewed in more than once phase and secondly, while all student interviews were individual, some staff interviews were conducted as group interviews [48].

**Table 1.** Data collected and analysed.

| Participants | Data | Method | Analysis | Research Phase |
|---|---|---|---|---|
| Students | Numerical (n. 45) <br> Free text | Survey (Appendix A) <br> Open survey items | Descriptive statistical <br> Deductive thematic | Phase 1 <br> Phase 1 |
| Students | Transcripts (n. 10) | Semi-structured interviews | Reflexive Thematic Analysis | Phases 2 and 3 |
| Academic staff | Transcripts (n. 5) | Semi-structured interviews | Reflexive Thematic Analysis | Phases 1, 2 and 3 |
| Professional staff | Transcripts (n. 4) | Semi-structured interviews | Reflexive Thematic Analysis | Phases 1 and 3 |
| Senior management staff | Transcripts (n. 2) | Semi-structured interviews | Reflexive Thematic Analysis | Phase 1 |
| Student Union Representatives | Transcripts (n. 2) | Semi-structured interviews | Reflexive Thematic Analysis | Phases 1 and 3 |

Conducting this research introduced specific ethical considerations, most importantly, maintaining student anonymity. The necessary approval was obtained prior to the evaluation from the institution where the main researcher was registered for a PhD. This was also confirmed by the relevant senior staff in the institution where the research took place. These ethics requirements, which are also British Educational Research Association (BERA) compliant [49], included the anonymisation of all survey data and the use of pseudonyms

for all interviewees. The partnership process itself was open to final-year student volunteers and no cap was imposed on the number of students who might be involved. Students received no payment for their involvement. After the process was completed and changes to the curriculum incorporated, reflections were gathered via interviews with students (all individual interviews) and staff (some individual and some group interviews). This paper extends that reflection and begins the process of reflection and reconstruction for planning the next cycle of curriculum review. The research gathered data regarding staff and students' desires for, and perceptions of, SaP in the ESD-focused curriculum review process. It uses the results of the survey data and interview data from which to draw insights into participant understandings and aspirations and identify factors that either inhibited or promoted the use of SaP in this context.

### 4.2. The Curriculum Review Process

It was the aim of all the authors to give students a real opportunity to influence the existing programme of study. Students have firsthand up-to-date experience of their learning journey and know things that staff do not [50]. So, it was seen as important to fully involve the students in the curriculum review as partners in the process. However, it was recognised that a goal of achieving this level of involvement was ambitious and may be hard to achieve in practice. In response to the invitation to participate, eight student volunteers came forward.

Upon agreement being reached to proceed, a schedule of intended actions and meetings was devised to incorporate the staff–student partnership within the process, which would culminate in the formal curriculum review event. Preliminary meetings were arranged with all cohorts of students to explain the process of curriculum review and to invite them to participate in the survey. At the same time, the students were introduced to some of the themes of review, including the concept of ESD and its relevance within legal education. This was carried out using a modified version of open-access resources for ESD curriculum design [6]. The survey was then sent to all students to assess perceptions of their existing level of control over the curriculum, their aspirations for change, what they hoped to gain from engagement with the process, their motivation for involvement, and what barriers to success they anticipated.

After initial discussions regarding format and timing, a meeting took place between students and all staff to discuss preliminary views on areas of current satisfaction and dissatisfaction within the curriculum, identifying some possible areas for change. This initial meeting was facilitated by a member of professional staff. The review progressed via correspondence and further meetings, both online and face-to-face, and with different configurations of participants over the following weeks. Students' views on the process itself were sought as the review progressed. Some adjustments were made to the schedule of meetings that had been planned at the outset, in particular, to accommodate discussion concerning additional student-led proposals for enhancement of the programme in the area of student wellbeing. This specific element of ESD, relating to the achievement of SDG 3, was seen by students as central to their desire for change. They argued strongly for the incorporation of curriculum content that related to wellbeing and its relevance to the study and practice of law, that is, for their experience as students and as future legal professionals. Awareness of the high levels of stress experienced by law students, academics, and legal professionals is increasing and discussions regarding how best to address this within the curriculum are ongoing [51,52]. In addition to student participation, other third-party stakeholders, including local employers and external examiners, were also consulted. Contributions from these sources tended to see ESD as climate- and environmentally focused and were positive regarding the explicit consideration of the role of law and legal regulation in these areas. For staff, though the inclusion of subject content relating to individual SDGs such as the environment (SDGs 13, 14, 15), gender equality (SDG 5), justice (SDG 16), or wellbeing (SDG 3) was seen as offering opportunities to enhance the curriculum, their main focus was on the incorporation of key ESD skills and competencies within appropriate

module learning outcomes. These competencies are taken from UNESCO guidance [3] and are set out in national HE policy [2] as the following: systems thinking; future thinking; critical thinking; strategic competency; normative competency; integrated problem solving; self-awareness; and collaboration competency.

As the deadline approached for submission of formal documentation drafted by staff, student participants continued to be involved, with comments sought on the draft proposals for change. The final written curriculum proposal was submitted with the support and agreement of all participants. The central proposals following consultation were to embed both ESD and specific wellbeing aspects into the programme and to introduce a more streamlined system of assessments. All eight of the ESD competencies are now reflected within relevant module learning outcomes. For example, a module regarding constitutional law was adjusted to incorporate the development of systems and future thinking in relation to changes in the UK's constitution including protection of both human and nature rights. Additional programme outcomes were included to cover ESD and wellbeing, designed to run as a thread through modules across all levels. These require development of the *knowledge, competencies, and ability to pursue sustainable visions of the future* and *understating of the importance of wellbeing and its impact on learning, employment, and personal and professional relationships*. A further specific example of a change that resulted directly from student requests was the reintroduction of timed exam assessments. To the surprise of staff, students felt the need for exams to prepare them for later post-graduate professional assessments. Staff therefore separately discussed and formulated draft proposals reflecting these views, which were then endorsed by students. The formal curriculum review event took place shortly afterwards, led by internal and external panel members and attended by law academics, two student representatives (who had been part of the SaP process), and a student reviewer. The curriculum proposal was approved unconditionally, noting collaboration with current students as an item of best practice.

Throughout the process, different motivations and priorities concerning ESD together with reservations regarding the benefits and use of a partnership process were discussed. As noted above, staff conceptions of ESD tended to focus on broader ESD subject knowledge and skills, while students' main focus was on wellbeing. All participants saw great value in the inclusion of diverse perspectives but highlighted the risks of increasing student power and diminishing staff expertise. These issues were captured in the research data and are discussed in more detail below.

## 5. Results

In this section, the results of the data analysis are considered. In the following section, they are then discussed in light of the SaP literature and theory regarding partnership as a form of student engagement.

The results of the survey of students were instructive. Students' perception of their existing control over the curriculum tended towards either ambivalence or negativity. Whilst 43.8% did feel that "tutors have control but are informed by student consultation or feedback", a further 21.9% felt that "students are consulted but have no actual influence" and 9.4% felt that there was no consultation at all. Only 9.4% responded positively that "students have influence through negotiation with tutors". When asked what level of control they ought to have however, 31.3% felt they should have influence through negotiation with tutors and a further 25% felt they should have influence over certain aspects of the curriculum. None felt it acceptable for there to be no consultation or influence. This indicates that students saw a more democratic model as desirable. We use the term democratic here in a deliberative, rather than directly representative, sense. This reflects the fact that partnership is not the same as student voice, but nevertheless values plurality, inclusivity, and participation. Notably, a large majority of respondents also felt that their views ought to be taken into account in the light of their tuition fees paid. This indicates that, at least in part, students understood their role to be as consumers.

As regards their motivation to be involved with the process, the most important factor identified was an altruistic one: the hope of benefitting future students. This can be seen as a perception on the part of students, rather than a definite outcome. It appeared to be based on the fact that the curriculum would be improved, but due to the fact that questions might be interpreted in various ways in surveys, a precise interpretation is not possible. Interestingly, this aspect was rated higher than gaining skills through experience or influencing the quality of the programme, which had been identified by staff as key benefits of students' participation. The least important of possible promoting factors for taking part was the idea that engagement was a policy requirement: students did not wish to see their involvement as a box-ticking exercise or their role as followers of institutional or political agendas. Nevertheless, views on barriers to the process indicated surprisingly little cynicism regarding the purpose of the invitation to participate, and most felt that their involvement would make a real difference to the outcome. The main discouraging factor was identified as time pressure and how any time necessary for the project might impact upon academic study and completion of assessments, as well as on outside work commitments.

Cook-Sather et al. [23] set out a number of positive outcomes of SaP for staff and students. For students, these include increased confidence, a stronger sense of identity, insight into staff pedagogy, and a contribution to the academic community. For staff, they include transformed thinking about teaching practices, a deeper understanding of student needs, and a reconceptualisation of students as colleagues. We now consider the benefits in terms of curriculum enhancement and the rewards for staff and students taking part in the process before moving on to discuss some of the risks perceived by participants.

The interview transcripts were analysed by one of the co-authors using NVivo and subjected to Reflexive Thematic Analysis [47]. This method of analysis allows for the generation of multifaceted themes across a whole dataset. This means that an overarching story that brings together multiple perspectives is represented. No group of participants is privileged, and theory, as understood by the researcher, informs, but does not dictate, theme generation. This is in contrast to other forms of thematic analysis which can be used to highlight predetermined topic areas for specific groups such as benefits or concerns. The three overarching themes generated were the following:

(a) The more voices the better—reflecting strong positive attitudes towards increasing the diversity of those involved in the review;
(b) It's not just box ticking—reflecting aspirations for partnership as a means by which to achieve personal, institutional, and societal outcomes but not policy compliance;
(c) There's a risk—reflecting concerns and understandings that might inhibit partnership.

The interview data showed that the majority of student participants were highly positive when reflecting on the experience. For example, one volunteer reflected, "*it's been brilliant, you know, we've talked, you've listened, you've talked, we've listened, and I think it's been a fair exchange of ideas and hopefully we've created something great*" (Lilly, phase 3). The advantages highlighted by Cook-Sather et al. [23] listed above were all evident in the data. This may be closely related to the strength of the altruistic motive within the student partners. This is a notable finding as altruism is not commonly reported as a promoting factor for SaP, although Noufou et al. [53] found that altruism was a "significant predictor" of students' willingness to mentor their peers. Participants reported satisfaction through giving back, both to the institution and also to future students. This aspect also appeared implicitly within the review process through their support for embedding wellbeing into the new curriculum. For example, another volunteer noted, "*I want to help people and this is like another opportunity where I can help people, and I think knowing that I've helped people. . . I found that quite rewarding*" (Nicki, phase 2).

For academic staff, the advantages outlined above were also evident, including a recalibration of their relationship with the students involved. However, within this research, the potential for problems in a shift towards a more collegiate relationship was also identified. For example, one academic explained, "*[Y]ou start thinking more in terms of colleague rather*

*than student, which you don't want to fully do when you're still marking their assessments and having to approve their request for extensions of time. There is an appropriate distance that you have to maintain to make sure that you're being fair to everyone*" (Alison, phase 2). This concern regarding an appropriate distance between staff and students meant that the extent of reconceptualisation as colleagues was partial and limited to interactions within the process. Other risks, identified by students and staff at all levels, appeared to relate to various risks in taking a partnership approach. For example, one senior manager warned,

"[W]*hat we don't want to do is open a massive can of worms for both staff and students... I think the key thing around that, is managing both student expectations on that, but also giving staff sufficient guidance. So they're clear, so they don't . . .feel they're under particular risk of getting it wrong*" (Rick, phase 1). Nevertheless, at the end of the process, there was recognition of the particular power of the student voice when combined with academic staff voices in adding support to the final proposal and enhancing its reception by third parties.

## 6. Discussion

In this section we discuss the results and consider the extent to which they align with, or challenge, current partnership theory. In addition, evaluative lessons learned are offered, not as a generalisable recipe for action, but as informative, context-dependent conclusions from which others might draw inferences for their own practice.

There has been some critique of the partnership literature which notes the positive bias in reporting with less consideration of more negative aspects of partnership processes [8]. In acknowledging the less positive aspects, we indicated above that several concerns were also raised by students and staff regarding risks within the review process. The first area of concern related to the rationale for the SaP approach. There were differences of opinion amongst staff as to the likely practical impact of student involvement upon the content of the review. While some were positive, others were wary of the impact on the process in terms of the time taken to conduct meetings and the impact on their ability to speak freely. Doubters nevertheless recognised that there might be rewards beyond the content of the curriculum proposal allowing students to take on an apprentice-style role, to achieve greater insight, enhancing their learning and thus aligning with the recognised advantages of SaP [23]. Tutors were also conscious of possible suspicion from students that their collaboration was being sought to satisfy an administrative requirement rather than there being any real intention to incorporate their ideas. This appeared to be justified by a student comment that, although they were happy with the outcome, at the beginning, "*it kind of felt like you kind of have to do it, you know, I mean get our input . . . but you didn't necessarily need to actually listen to it*" (Carrie, phase 3). In fact, there was no political agenda mandating the process in this case. It was therefore agreed that clear explanation and reassurance regarding staff motivation were key to instilling trust within the process.

Secondly, though it was recognised that diverse and more democratic engagement and the inclusion of a wide variety of perspectives ought to lead to a better outcome, it was clear that priorities regarding change to the curriculum were often different, especially between staff and students. Staff voiced concern regarding the management of student expectations, but in fact, both students and staff reflected on the value of accommodating divergent priorities in achieving the final outcome. There were also clear differences regarding the practicalities of the project in terms of the means of contact (either online or face-to-face or a combination), with different participants voicing contradictory views regarding how to generate both the best quality and quantity of discussion. Such differences are characteristic of a complex environment. Though these differences were not correlated with student or staff roles, this nevertheless required considerate management based on trust, compromise, and discussion.

The final area of concern related to the perceived risks of involving students in an "equal partnership" [23]. There was discussion concerning the importance of defining what was meant by "partnership"—a word with some specific definitions within the legal context—to ensure all participants were clear about the intended purpose of their

involvement. Some staff were concerned not to raise students' expectations of proposing changes which if not later included could be a cause of dissatisfaction or even complaint. There was discomfort from staff regarding expectations of equal partnership, because it was the academic team's ultimate responsibility to put forward a good proposal. Both staff and students highlighted broader systemic issues, including the need to comply with a variety of university and professional requirements, restrictions, and guidance of which students would mostly be unaware. In addition, the idea that including students throughout the formal review process could challenge and perhaps alter staff–student relationships was raised as both a potential benefit and a potential problem. Beneficial elements included increased democracy in decision making and additional student learning. For example, one student volunteer reflected, "*diverse opinion[s] should be involved, I think it was really, really beneficial*" (Hazel, phase 3). Problematic elements concerned increasing the relative power of students and therefore diminishing the power of staff. Participants appeared to see power as zero-sum and thus the highest level of participation by students, i.e., student control, as a potential though extreme outcome of the process [54]. One academic, anticipating the role of student partners, stated, "*[m]y concern and fear would be if . . . they weren't aware of why they're involved and what their role of involvement is. If that was misinterpreted to a power issue, an issue that I have power and I can tell you what to do . . . I think that would be a heavy risk of dissatisfaction and problematic review*" (Ben, phase 1). However, as Ashwin and McVitty [55] point out, more student engagement is not necessarily always better (p. 356). Seeing power not in zero-sum terms but instead as inherent in a community of inquiry, comprising students and staff, reflects the value placed on democracy and diverse viewpoints but avoids the risks perceived in a rebalancing of power relationships. We found that the learning community at the centre of partnership activities in the SaP model (Figure 1) can usefully be seen as a community of inquiry. Within HE, the concept of a learning community can take many forms and learning communities are used in various ways [56,57]. We agree with Lower [58,59], who suggests the use of a community of inquiry in processes of change in curriculum where staff and students act as partners in knowledge production. The community of inquiry model focusses on empowerment through collaboration, situating power within the learning community of staff and students [56,60]. The associated theoretical community of inquiry framework [61] was designed for use in blended online and face-to-face environments. The framework uses a practical inquiry model [62] comprising four phases: problem identification as a trigger for inquiry, followed by exploration of the subject including reflection, leading to integration and application of newfound knowledge towards resolution. In this process the triggering event was the question of how to incorporate ESD in the curriculum, exploration involved students and staff considering alternative proposals, and integration of their priorities finally led to resolution in a revised curriculum.

Developmental evaluation promotes the articulation of lessons learned based on high-quality information and evidence. Having discussed some of the perceived risks and rewards of the process, while highlighting the lack of direct generalisability of this study, we set out below some of the key lessons learned. Although student volunteers were involved in evaluating the process, the lessons below were developed by academic staff.

### 6.1. Getting Academic Staff Onboard Early

We found that early focus should be directed towards gaining the support and confidence of academic staff involved in taking an innovative approach to the review. An exploratory meeting creates the opportunity to explain the rationale, air views, and identify and discuss reservations. Practicalities including which cohort(s) of students to involve, and the nature and extent of their involvement can be decided in this planning stage, alongside the preparation of a timeline and project plan. Consideration could be given to including final-year students as a priority. This cohort have had the longest experience of the programme and may have developed an altruistic sense of wishing to share their insights and experience to benefit their programme and its future students.

### 6.2. Seeking Volunteers and Establishing Trust in the Process

The process of seeking volunteers requires careful planning. Potential participants need sufficient context of the procedure and what is being requested of them. Reassurance should be given regarding likely concerns, which may include doubts about the nature of the proposed partnership, the value of their contribution, and their likely time commitment. It will be important to dispel the idea of the collaboration as an administrative requirement of the process and instead portray a more positive aim where diverse views are highly valued. Placing emphasis on positive rewards for students is a helpful strategy: they can gain useful professional and employability skills through liaison with professionals, discussion, negotiation, and teamwork; they will gain personal learning through insight into the process behind the scenes; and they may achieve great personal satisfaction from helping the next generation of students by being involved in the process of co-creation.

### 6.3. Encouraging Volunteers to Speak Freely

It can be anticipated that despite being willing to participate, some students may still feel unsure about expressing views openly in meetings which are subject to a different dynamic than that previously experienced in the lecture theatre or classroom. Once volunteers have come forward, it will be important to expand focus on the nature of the "partnership". Within the SaP process, it is intended that students "have the opportunity to contribute equally, although not necessarily in the same ways" [23]. It is suggested that at a first meeting with volunteers, the meaning and extent of the partnership should be discussed and explained fully. Students may be concerned that within the existing power relationship, their views could be disregarded or even that opinions critical of the current programme might cause offence. The perceived power of academics should be reframed as their responsibility for the programme, which includes attaching significant respect and value to student contributions. Barriers to expression might also be present due to a sense of a lack of expertise. Reassurance can be given that it is recognised that the student perspective is different and that while academics and students have different perceptions, responsibilities, and areas of expertise, student participants have valuable observations to offer and a diversity of views within a discussion will create a firmer foundation for debate and change.

### 6.4. Being Open to Diverse Motivations and Unexpected Outcomes

As discussed above, participants in the curriculum review described here had different motivations for engaging in the process. Furthermore, none of these completely aligned with all the stated proposes for using SaP from the relevant literature. Despite this, the process was seen as valuable and the outcome, as having been enhanced. Overall, while the academic staff's desire to incorporate ESD was more influenced by relevant HE guidance and broader policy, students' motivations appeared to be based on personal experience and related to the effect of change on their peers. Bringing these together created an unexpected but arguably better outcome in this case.

### 6.5. Seeing Power as a Tool to Be Used within a Community of Inquiry

By avoiding presentations of power as being held or relinquished by either staff or students but instead used together in collaboration, some of the perceived risks of partnership practice might be avoided to some extent. Presenting collaboration as enabling the empowerment of a community of inquiry may be useful in motivating participation while emphasising joint action. This has the advantage of focusing on potential action rather than highlighting risks such as those raised in this example by staff and students of complaints, inappropriate use of expertise, or loss of power despite continuing responsibility.

## 7. Conclusions

We have argued that collaboration and partnership are fundamental to ESD. Collaborative competency is one of eight competencies highlighted within ESD. Furthermore,

SDG 4, education, and SDG 17, partnership, highlight the significance of education and partnerships in achieving the goals. Addressing this, we reported on a developmental evaluation undertaken within a small cohort of law students during which students worked in partnership with staff to incorporate ESD in the law curriculum in ways that reflected participant priorities.

This highlighted the different motivations of staff and students within a complex environment, but also the perceived value of their diverse perspectives. The rewards of the process were seen by participants to encompass both their own learning and development through the process and the outcome in the form of an enhanced curriculum. However, risks were also highlighted by students and staff, some of which appeared to arise from the prospect of a rebalancing of the power relationship between them. It appeared therefore, that participant understanding and aspirations could act to both promote and inhibit working in partnership. Therefore, mitigation of inhibiting factors such as concerns regarding risk or power might promote use of partnership in practice. Concerns regarding power can be linked to systemic issues including the concept of students as consumers. We suggest that such concerns can be mitigated by focusing on the empowerment of a collaborative community of inquiry to address the problem of curriculum change. Further research in this area might usefully test the relevance of the Community of Inquiry framework to SaP activities. When undertaken in a developmental way, the innovation of co-creating curricula can address current challenges in a timely and iterative fashion. We found this to be one of the main advantages of a developmental evaluative methodology, which was also sufficiently flexible to accommodate the methods of data collection that were felt useful and appropriate in this context. However, developmental evaluation is a relatively new, and therefore untested, approach in educational research. In other contexts, where the timing of institutional processes is different, there may be tensions between these processes, data analysis, and optimal evaluative timeframes [63]. We acknowledge the limitations of small-scale context-dependent research which is not generalisable. In addition, our consideration of the community of inquiry framework theory was necessarily brief and this could usefully be the subject of further research in similar contexts. However, we offer our experience and recommendations based on the process described for others to consider when planning their own curricula changes.

**Author Contributions:** Conceptualisation, F.A.B.; methodology, F.A.B. assisted by F.M.B.; formal analysis, F.A.B.; resources and organisation, F.A.B., F.M.B. and D.R.; writing—original draft preparation, F.A.B., F.M.B., D.R. and K.A.A.G.; writing—review and editing, F.A.B., F.M.B., D.R. and K.A.A.G. All authors have read and agreed to the published version of the manuscript.

**Funding:** This research received no external funding.

**Institutional Review Board Statement:** All subjects gave their informed consent for inclusion before they participated in this study. This study was conducted in accordance with the Declaration of Helsinki, and the protocol was approved by the Ethics Committee of Lancaster University, Faculty of Arts and Social Sciences and Management School Research Ethics Committee (FASS-LUMS REC-FAB-2-12-2021).

**Informed Consent Statement:** Informed consent was obtained from all subjects involved in the study.

**Data Availability Statement:** The data presented in this study are available on request from the corresponding author. The data are not publicly available due to the possibility that participants may be identifiable.

**Conflicts of Interest:** The authors declare no conflicts of interest.

**Appendix A**

## Page 1

What is your year group?

Are you, or have you been in the past, a student representative for your year group?

○ Yes - in the past
○ Yes - currently
○ No

Thinking about the LLB as a whole, please indicate the extent to which you agree that a review of the LLB curriculum should consider changes to the following:

Please don't select more than 1 answer(s) per row.

|  | strongly agree | agree | neither agree nor disagree | disagree | strongly disagree |
|---|---|---|---|---|---|
| Academic legal knowledge | ☐ | ☐ | ☐ | ☐ | ☐ |
| Skills development | ☐ | ☐ | ☐ | ☐ | ☐ |
| Methods of teaching (lectures, seminars, workshops) | ☐ | ☐ | ☐ | ☐ | ☐ |
| Delivery format (face to face, blended, online) | ☐ | ☐ | ☐ | ☐ | ☐ |

3 / 10

**Figure A1.** *Cont.*

| | | | | | |
|---|---|---|---|---|---|
| The order in which modules are studied | ☐ | ☐ | ☐ | ☐ | ☐ |
| Methods of assessment | ☐ | ☐ | ☐ | ☐ | ☐ |
| Preparation for legal practice or other employment | ☐ | ☐ | ☐ | ☐ | ☐ |
| Integration of a sustainable development perspective | ☐ | ☐ | ☐ | ☐ | ☐ |
| Integration of an equality, diversity and inclusion perspective | ☐ | ☐ | ☐ | ☐ | ☐ |

Are there any other aspects of the LLB not referred to above that you consider important to include in a review of the curriculum?

The curriculum review process will take place between January and May, and may include a training event, online meetings including staff and students, reviewing draft documents and helping to make decisions. Thinking about this, please indicate the extent to which you agree with the following statements:

Please don't select more than 1 answer(s) per row.

| | strongly agree | agree | neither agree nor disagree | disagree | strongly disagree |
|---|---|---|---|---|---|

4 / 10

**Figure A1.** *Cont.*

| | | | | | |
|---|---|---|---|---|---|
| Students should be involved in the curriculum review process | ☐ | ☐ | ☐ | ☐ | ☐ |
| Only the confident students would take part | ☐ | ☐ | ☐ | ☐ | ☐ |
| It would be a token gesture to comply with university policies | ☐ | ☐ | ☐ | ☐ | ☐ |
| It would enable students to learn from staff | ☐ | ☐ | ☐ | ☐ | ☐ |
| It could give students too much power | ☐ | ☐ | ☐ | ☐ | ☐ |
| It would enable students to work with staff towards improving the curriculum | ☐ | ☐ | ☐ | ☐ | ☐ |
| It would enable students to contribute to decision making in the university | ☐ | ☐ | ☐ | ☐ | ☐ |
| It would improve student satisfaction | ☐ | ☐ | ☐ | ☐ | ☐ |
| The university may want to do it for marketing reasons | ☐ | ☐ | ☐ | ☐ | ☐ |
| It would be a more democratic process if students were involved | ☐ | ☐ | ☐ | ☐ | ☐ |

5 / 10

**Figure A1.** *Cont.*

| | | | | | |
|---|---|---|---|---|---|
| Volunteers from year 1 and year 2 should also be included | ☐ | ☐ | ☐ | ☐ | ☐ |
| Students might use it as an excuse to make complaints | ☐ | ☐ | ☐ | ☐ | ☐ |
| It is important for student views to be taken into account in view of the fees they pay | ☐ | ☐ | ☐ | ☐ | ☐ |
| It would not make much difference to the outcome | ☐ | ☐ | ☐ | ☐ | ☐ |
| The views of participating students will not be representative of other students | ☐ | ☐ | ☐ | ☐ | ☐ |
| It would give students more power | ☐ | ☐ | ☐ | ☐ | ☐ |
| It would result in a better curriculum | ☐ | ☐ | ☐ | ☐ | ☐ |

How much control do you think students *currently* have over the curriculum? Please indicate the level you think best fits your view.

○ Students are in control of the curriculum
○ Students have influence through negotiation with tutors
○ Students have influence over certain limited aspects
○ Tutors have control but are informed by student consultation or feedback
○ Tutors have control with limited student choices

6 / 10

**Figure A1.** *Cont.*

○ Students are consulted but have no actual influence

○ No consultation of students takes place

○ Other

If you selected Other, please specify:

How much control do you think students **should** have over the curriculum? Please indicate the level you think best fits with your view.

○ Students should be in control of the curriculum

○ Students should have influence through negotiation with tutors

○ Students should have influence over certain limited aspects

○ Tutors should have control but be informed by student consultation or feedback

○ Tutors should have control with limited student choices

○ Students should be consulted but have no actual influence

○ No consultation of students is needed

○ Other

If you selected Other, please specify:

If your year group was represented in the review process meetings, how likely would you be to volunteer to take part in meetings and to review documents as part of the curriculum review?

☐ Very likely

7 / 10

**Figure A1.** *Cont.*

☐ Likely

☐ Unsure

☐ Unlikely

☐ Very unlikely

☐ I have already volunteered

If you would like to explain your answer to this question please do so here.

If you were able to take part in the review, what would motivate you to volunteer? Please indicate all that apply and their importance to you.

Please don't select more than 1 answer(s) per row.

| | Very important | Quite important | Not important | Not at all relevant |
|---|---|---|---|---|
| To learn about the process | ☐ | ☐ | ☐ | ☐ |
| To gain new skills | ☐ | ☐ | ☐ | ☐ |
| To comply with university policies about including students | ☐ | ☐ | ☐ | ☐ |
| To contribute to the discussion | ☐ | ☐ | ☐ | ☐ |
| To influence the quality of the degree | ☐ | ☐ | ☐ | ☐ |
| To put on my CV | ☐ | ☐ | ☐ | ☐ |
| To benefit future students | ☐ | ☐ | ☐ | ☐ |
| To help tutors see the student perspective | ☐ | ☐ | ☐ | ☐ |
| To make the process more democratic | ☐ | ☐ | ☐ | ☐ |

8 / 10

**Figure A1.** *Cont.*

Are there any other reasons not listed above?

Are there any further comments or views you would like to add?

9 / 10

**Figure A1.** The survey instrument, showing all questions but omitting initial ethics and consent information.

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
