# Peer review of "Exploring Staff–Student Partnership in Curriculum Design"

_education, doi:10.3390/educsci14010061_

Round 1

Reviewer 1 Report

Comments and Suggestions for Authors

The topic of the manuscript is relevant and fits within the scope of the journal Education Sciences. The manuscript, entitled "A Collaborative Curriculum: Navigating the Risks and Rewards of Staff-student Partnership in Curriculum Design' describes an example of collaborative curriculum change using the Students as Partners (SaP) model to update an undergraduate programme to include Education for Sustainable Development.

In this way, the manuscript aligns with the aims of the special issue "Teacher Narratives and Narratives of Teaching: Inquiry into Lived Experiences".

The overall assessment of the paper is positive, but some efforts are needed to make it acceptable.

I agree with the methodological approach adopted, but more reflection and justification of its appropriateness is needed. The manuscript can be strengthened by a more detailed presentation of the methodology. The research methods and procedures are not clearly and sufficiently described to allow other researchers to replicate them. Some further points for improvement

- It is necessary to explicitly describe the validation process of the main data collection instruments.

- It would be enriching to share the survey and interview scripts and a summary table of the content analysis.

- It would be important to provide a more detailed characterisation of the participants.

Best wishes for the rewriting process.

Reviewer 2 Report

Comments and Suggestions for Authors

The article is very clearly expressed and is well organised. I have very few concerns about either its style or its content. The authors have a lighter touch with commas than I do, but that is a stylistic choice. I noticed only one typo: 'as' on line 312.

There are, at various points, references to 'benefits', 'rewards', 'positive outcomes' and 'drivers'. I think the authors should consider whether these terms are used consistently and to what extent they overlap. Are 'rewards' the benefits to the participants and 'benefits' apply to current and future students? Take lines 435 to 450, for example. Is altruism a driver or an outcome? 

A claim is made concerning democracy. I'm not sure that the participation described necessarily amounts to democracy. Doesn't it depend on to what extent student voice affects actual decisions? The article could be clearer about exactly how the students' contributions impacted actual decision making and how decisions were reached. More generally, the article would benefit from some precise examples. Were any of the discussions recorded or transcribed?

Another claim is that the students' participation will have helped future generations of students (lines 555/6).  How exactly? Is this a reference to the 'enhanced' law curriculum, to future students' understanding of sustainability, or both? Is it a perception or a definite outcome?

Overall, I found the article interesting and engaging. It is a appropriately scholarly.  On reflection, it is perhaps surprising that we do not hear from the students themselves. There is scope here for the inclusion of some dialogue, with accompanying commentary or analysis. Were the student participants consulted in any way in the preparation of the article?

Reviewer 3 Report

Comments and Suggestions for Authors

The conclusion and discussion were relevant; however, it would be interesting to the reader to know what changes occurred in the law program due to this unique process of Education for Sustainable Development (ESD).

Reviewer 4 Report

Comments and Suggestions for Authors

I congratulate the authors of this article for their choice of research topic and its rigorous treatment. The arguments underpinning this case study are solid and very well constructed. The whole text is coherently presented and very carefully written to facilitate comprehension. I highlight the direct link between the selected method (SaP) for collaborative curriculum development and the competences involved in ESD. This linkage is intelligent and brings more value to the process of curriculum improvement.

The results obtained are well presented and consistent with the objectives of the study. The conclusions are well developed and their contributions are valuable, even considering the limitations expressed by the authors themselves regarding the size of the sample. I would like to encourage researchers to continue their studies using Garrison's "Community of Inquiry" model. I consider that its introduction by the authors is a good idea and that, in the future, this model can improve the understanding of this type of collaborative practices.

In order to improve the article, it is recommended that some extracts from the transcripts of interviews or meetings be included. Their purpose is to provide data to support the results and reinforce the arguments offered in the conclusions. I reiterate my congratulations to the authors of the study.

Round 2

Reviewer 1 Report

Comments and Suggestions for Authors

All questions arisen have been addressed, answered and duly implemented by authors. Paper has won in clarity and quality. Now the paper is suitable for publication in its current form.